# Monitoring Complex Integrated Crop–Livestock Systems at Regional Scale in Brazil: A Big Earth Observation Data Approach

Patrick Calvano Kuchler [1,2,*], Margareth Simões [1,3], Rodrigo Ferraz [3], Damien Arvor [4], Pedro Luiz Oliveira de Almeida Machado [5], Marcos Rosa [6], Raffaele Gaetano [2,7] and Agnès Bégué [2,7]

1   Department of Computer Engineering, Rio de Janeiro State University (UERJ/FEN/DESC/PPGMA), Rua São Francisco Xavier, 524, 5031 D, Maracanã, Rio de Janeiro 20550-900, Brazil; margareth.simoes@embrapa.br
2   TETIS, University Montpellier, AgroParisTech, CIRAD, CNRS, INRAE 648 Rue Jean François Breton, 34090 Montpellier, France; raffaele.gaetano@cirad.fr (R.G.); agnes.begue@cirad.fr (A.B.)
3   Embrapa Solos, Rua Jardim Botânico 1024, Rio de Janeiro 22460-000, Brazil; rodrigo.demonte@embrapa.br
4   CNRS, UMR 6554 LETG, Université Rennes 2, 35043 Rennes, France; damien.arvor@univ-rennes2.fr
5   Embrapa Arroz e Feijão, Rodovia GO-462, km 12, Santo Antonio de Goias 75375-000, Brazil; pedro.machado@embrapa.br
6   Department of Geography, PPGM Universidade Estadual de Feira de Santana, Novo Horizonte 44036-900, Brazil; mrosa@arcplan.com.br
7   CIRAD, UMR TETIS, 34398 Montpellier, France
*   Correspondence: geocalvano@gmail.com; Tel.: +55-021-98010-6336

**Abstract:** Due to different combinations of agriculture, livestock and forestry managed by rotation, succession and intercropping practices, integrated agriculture production systems such as integrated crop–livestock systems (iCL) constitute a very complex target and a challenge for automatic mapping of cropping practices based on remote sensing data. The overall objective of this study was to develop a classification strategy for the annual mapping of integrated Crop–Livestock systems (iCL) at a regional scale. This strategy was designed and tested in the six agro-climatic regions of Mato Grosso, the largest Brazilian soybean producer state, using MODIS satellite time-series images acquired between 2012 and 2019, ground data with heterogeneous distribution in space and time and a Random Forest classifier. The results showed that: 1. the use of unbalanced training samples with a class composition close to the real one was the right classifier training strategy; 2. the use of a single training database (pooling samples from different years and regions) to classify each region and year individually proved to be robust enough to provide similar classification accuracies in comparison to those based on the use of a database acquired for each region and for each year. The final hierarchical classification overall accuracy was 0.89 for Level 1, the cropping pattern level (single and double crops DC); 0.84 for Level 2, the DC category level (integrated system iCL soy-pasture/brachiaria, soy-cotton and soy-cereal); 0.77 for Level 3, the iCL level (iCL1 soy-pasture and iCL2 soy-pasture mixed with corn). The F-scores for DC, iCL and iCL1 cropping systems presented high accuracy (0.89, 0.85 and 0.84), while iCL2 was more difficult to classify (0.63). This approach will next be applied across the entire Brazilian soybean corridor, leading to an operational tool for monitoring the adoption of sustainable intensification practices recognized by Brazil's Agriculture Low Carbon Plan (ABC PLAN).

**Keywords:** cropping systems; double cropping; sustainable agriculture; satellite image time series; MODIS; machine learning; big data; hierarchical classification; training sample designs; samples balancing

## 1. Introduction

Brazil is currently the world's largest producer of soy and meat [1]. In the last six decades, the country has doubled its agricultural area, mainly due to the inclusion

of new production areas in the Amazon and Cerrado biomes [2,3]. However, nowadays, it is possible to observe a clear land-use intensification process based on the adoption of integrated production systems, notably in the agriculture areas related to the grain production chain. The Crop–Livestock and Forestry Integration (iCLF) is one of the main land use axes of the Sectoral Plan for Mitigation of Climate Change for Agriculture, the so-called ABC Plan (Low Carbon Agriculture; [4]). It is a valuable strategy to increase production in a more sustainable way and without the need to expand the production area, as it becomes possible to efficiently produce different crops, grains, forest essences and animal production in the same area [5,6]. Because of these agro-environmental benefits [7,8], integrated systems have been widely adopted in Brazil [7,9,10].

The integrated crop–livestock–forest (iCLF) and integrated crop–livestock (iCL) are particular sequential systems that consist of agricultural production (i.e., grain crops and livestock), intercropped or in rotation grain crops with forage grasses, or in different parcels within a farm. Different spatio-temporal combinations of varied crops in agricultural production systems have been recognized as a strategy that contributes to increase the agroecological efficiency by providing more efficient use of nutrients and greater incorporation of soil organic carbon. Particularly, the introduction of pasture in the integrated systems intensifies this process due to the greater grass biomass production. Grain yields of crops grown in rotation with pasture are generally higher than crops grown in non-grazed fields [11] and, on the other hand, the conversion of low-yield extensive pastures to iCLs increases stocks soil carbon and forage productivity [12].

Stimulated by sectoral public policies, such as the ABC Plan, the monitoring the iCLs expansion area, on a national scale becomes extremely relevant for the sectoral management from the Agriculture Ministry of Brazil, which needs information to assess the level of adoption and the impact of these public policies. However, so far, the estimate of the implementation of these systems is made by surveys that are costly in time and resources, which is mostly carried out through interviews with cooperatives and producers [13,14]. Therefore, the development of monitoring systems that can automatically provide, updated information is strongly required.

Earth observation (EO) data have been widely used to map agricultural patterns or crop types and to monitor land use and cover changes over large areas; however, the detection of crop practices or production systems still constitutes a challenge in remote sensing. There are only a few examples of crop systems mapping in the specialized literature; they are, in fact, mainly exploratory studies, carried out at the local level, focusing only on certain practices [15]. Furthermore, Câmara et al. [16] point out that, despite the large volume of Earth Observation data available, only a small part is used for scientific research or operational applications. The results published in most cases are based on experiments carried out with limited datasets referring to small field areas and, therefore, are not properly tested and validated to support sectoral public policies.

However, the cropping practices' diversity, in particular the integrated crop–livestock systems, bring a further considerable complexity for land monitoring approaches based on remote sensing data. First, considering the properties and the temporal variability of these complex production systems, it is necessary to be able to capture the dynamics of land, and in particular, of sequential cropping, through the analysis of Satellite Image Time Series (SITS) [15,17]. As discussed by Gómez et al. [18] in their review, the vegetation indices (VI) extracted from the time series of images allow the reconstruction of the entire phenological cycle of the vegetation, which allows a better understanding of the multiple cropping pattern. For example, the MODIS (MODerate resolution Imaging Spectroradiometer) satellite has been successfully used for mapping simple and sequential crops over large areas, particularly in the United States, e.g., [13,14,19,20], China, e.g., [21,22] and Brazil [23–29]. In Brazil, Manabe et al. [30] and Kuchler et al. [31] carried out studies focused on the detection of iCL in small regions of Mato Grosso state. In both studies, good accuracy is presented, but the spatial extent of the final product is limited and does not correspond to the needs of the policy-makers. At the opposite, [32] mapped the soybean-pasture system over the

entire state of Mato Grosso but used only 20 samples referring to a small region, which made it impossible to correctly assess the accuracy classification.

New information technologies (Big Data approach) and satellite data clouds-processing platforms have allowed massive data processing, enabling solutions for large-scale land-use monitoring. As an example, we can mention the Google Earth Engine (GEE) platform, which allows, from a programming interface through an API (application programming interface), to access a petabyte catalog of satellite images and use Google's infrastructure to process them [33]. This way, spatial data processing provides an environment that makes it possible to compensate for the technical limitations of image processing over large areas. As shown in the meta-analysis published by [34], GEE is increasingly used worldwide, mainly for the mapping of agricultural areas, but a few studies are dedicated to mapping production systems, such as those based on sequential crops. An important initiative in Brazil that uses GEE for annual land-use monitoring is the Brazilian Project for Annual Mapping of Land Use and Occupation (MapBiomas), which has been mapping the entire national territory using Landsat images with 30 m spatial resolution for the last 33 years [35]. MapBiomas annually releases new maps versions that sometimes include new land use classes or improvements in image processing methods. The hierarchical nomenclature includes classes such as "soybean", "sugar cane" and "other annual crops" but does not currently include information about complex systems, including sequential crops and iCL practices.

In this context, the motivation of this study was to explore the possibility of using remote sensing data in order to contribute to the establishment of a methodological protocol for monitoring integrated systems in Brazil. Then, the main objective of this work was to develop a methodological approach for mapping the integrated crop–livestock systems in the state of Mato Grosso, Brazil. Secondarily, the study aimed to evaluate the strategy for building the training database in two main points of investigation: (i) identify a reliable sampling strategy when targeting a marked minority class, such as iCL; (ii) evaluate the generalization of classification models over areas and cropping seasons that were not covered by field data collection. The classification strategy was applied for seven agricultural years between 2012–2013 and 2018–2019.

## 2. Materials and Methods

### 2.1. Study Area

The methodology was applied in the state of Mato Grosso (MT), Brazil, which is approximately 903,357 km$^2$ (Figure 1a).

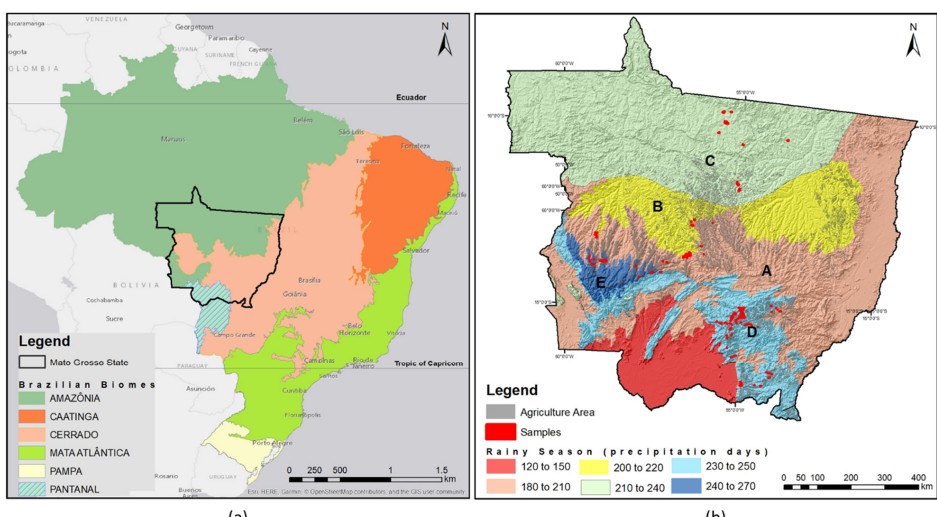

**Figure 1.** Study area: (**a**) Location of Mato Grosso state in Brazil and biomes map; (**b**) Agroclimatic maps of Mato Grosso state established in a previous study [36]; Cropland extent in 2017 is indicated in grey mapped by Camara et al. [27], and the ground samples are indicated in red (2018/2019).

### 2.1.1. Mato Grosso Environment

Mato Grosso contains three distinct biomes—the Amazon forest in the northern part of the state, the Cerrado savannah in central Mato Grosso and the Pantanal wetland in the southwest of the state (Figure 1a)—and six climatic zones (Figure 1b). These zones have been labeled in terms of five agroclimatic zones of agricultural potential (from low to excellent) where the zone with the lowest rainfall (between 120 and 150 mm) was not included according to the ecological-economic zoning of the state [36].

The predominant climate types are tropical, super-humid monsoon climate (with an annual average of 2000 mm) and tropical climate (with summer rains and dry winters with precipitation above the annual average of 1500 mm [37]). It still has an interannual climate variability (Figure 2) due to the phenomena of the South Atlantic Convergence Zone (SACZ), the Intertropical Convergence Zone (ITCZ) and the "ENSO" (El-Niño—Southern Oscillation) phenomena.

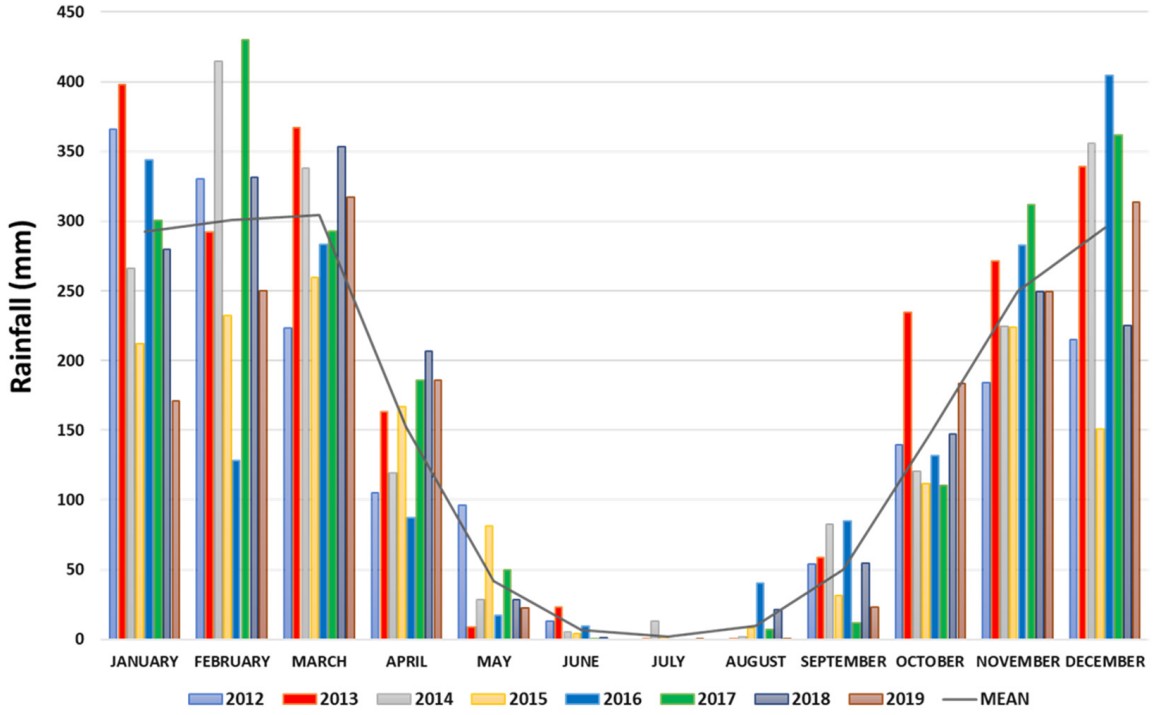

**Figure 2.** Histogram depicting monthly rainfall distribution for Mato Grosso in different years.

### 2.1.2. Mato Grosso Agriculture

Mato Grosso is one of the main national beef and soybean producers [38]. A total of 23.2% of the state is covered by pasture and 12% by agriculture [39] (Figure 1b).

According to official statistics from the Brazilian institute of geography and statistics (IBGE—*Instituto Brasileiro de Geografia e Estatística*) [40], in 2019, 95% of the state's agricultural areas were occupied with three main crops: soybeans (58.47%), corn (30.22%) and cotton (6.65%), normally cultivated in succession in the same crop year (double cropping systems). According to SEEG (*Sistema de Estimativas de Emissões e Remoções de Gases de Efeito Estufa*) [41], Mato Grosso has been, for the last 10 years, the Brazilian state with the highest Greenhouse Gases (GHG) emission rate in the agricultural sector (73% of livestock emissions, 6% from fertilizer use, 6% from soybean cultivation, 15% from other sources). To maintain high grain productivity and, at the same time, contribute to the mitigation of GHG emissions [7,42–45] crop–livestock–forest integration (iCLF) and crop–livestock integration (iCL) systems were particularly encouraged in the state. As a result, the state of Mato Grosso has the second largest area of integrated systems implemented in the country, of which more than 80% are integrated crop–livestock (iCL), with 1.5 Mha implemented for 2015/2016 [14].

Based on the field data collection reported by [46], two main strategies of the iCL systems are found in the state:

1.  The multi-annual iCL systems: These are systems based on multi-annual crop-pasture rotation. In this strategy, the farmer grows grains crops on the same plot for a few years, then introduces pasture for the cattle that remain for a few more years and then goes back to agriculture crops, and so on. This type of integration is generally adopted by farmers whose focus is both agriculture and livestock. It also constitutes a profitable pasture reform strategy used by ranchers with the aim of recovering degraded pastures and/or increasing the general productivity of livestock activity because the residual crop fertility contributes to restoring soil quality [47,48].

2.  Annual iCL systems: These are systems based on the succession of crops with pastures in the same agricultural year. Normally, in the studied region, soybean is cultivated in mid-summer and, depending on the duration of the rainy season, a second crop (corn, millet, sorghum) can be cultivated before planting a forage species to form pasture for cattle grazing. The remaining forage biomass can be yet used as mulch for planting the next crop in a no-tillage system. There are different strategies for the introduction of the forage species (*brachiaria*) in the iCL systems. In the regions where just one crop can be grown, due to the climatic restrictions, forage is normally introduced by overseeding after the soybean harvest. While, in regions where double cropping is possible, the forage is initially intercropped with the second crop, which is usually corn or another cereal. In this case, the corn grows first and is harvested before the pasture is fully developed, and after that, the pasture can fully develop.

However, it is observed that in both modalities of iCL, the introduction of forage occurs in the same way after the summer harvest. In the multi-annual case, the pasture for livestock is maintained for a few years, while in the annual case, grazing remains only until the new harvest. Therefore, to detect iCL systems, it is sufficient to evaluate the crop's phenological profile in succession at the same agricultural year: (i) 1st summer crop (soybean) + 2nd summer crop (corn/millet or sorghum) + pasture (brachiaria/or other forage species), for regions that allow double cultivation; (ii) 1st summer crop (soybean) + pasture (Brachiaria or other forage) for those regions whose rainy season is too short to sustain two crops at the same time. We focused on the detection of iCL systems considering the spectral profile characterization of the phenological cycles of the different cultures in succession over one cropping season, making it possible to separate the two aforementioned cases.

The results of the farms investigated during this work show that in Mato Grosso, integrated systems based on crop rotation also practice the integration of agriculture and livestock in sequential crops. Thus, the iCL system considered in this work refers to *boi safrinha*, which comes in two forms of sequential crop: soybean then pasture, and soybean then corn/pasture association (which is extended by pasture alone after the corn harvest).

### 3. Data

*3.1. MODIS Time Series*

Over the Mato Grosso state, 16-day composite MODIS Vegetation Indices and Spectral Bands time series were acquired at a 250 m spatial resolution (MOD13Q1 product; [49] for seven cropping years (October 2012–October 2019). Six tiles are needed to cover the entire state.

The MODIS time series of vegetation indices was not pre-processed, as recommended by Picoli et al. [27] and Chen et al. [32] for tropical areas and confirmed by our previous work [31], where we obtained the best cropping system classification accuracy in a northern Mato Grosso study site, including the integrated crop–livestock system class, using raw vegetation indices. It has also been shown that in that region, data filtering, such as Savitsky–Golay processing, on the one hand, reduces atmospheric interference in the time series, but, on the other hand, filters away important information, thus hindering the discrimination of classes with similar time profiles.

*3.2. Ground Data Collection*

The availability of ground datasets, up-to-date and spatially representative, remains a huge challenge for mapping agricultural land use on a large scale [15,50,51]. To get a significant number of field samples, it is often necessary to implement several data collection strategies. To label "stable" classes that have little intra and inter-annual variation (perennial crops or natural wooded vegetation), we can base ourselves on the visual interpretation of very high spatial resolution images to define learning areas [35,52]. For classes which, on the contrary, have a marked seasonal and annual cycle (annual crops), in situ data collection remains the most reliable method for building the "ground truth" database [53].

Parente et al. [54,55] use a combination of data collected by field campaigns and photo-interpreted data in order to map pastures. In the case of sequential cropping systems, the datasets are generally derived from interviews with producers and cooperatives, supplemented by data collected in the field [23,26,27,30,32]. The identification of these systems remains very difficult by photo-interpretation, even for specialists, because they are very dynamic and diverse systems, with different classes of pasture and annual crops (soybeans, cotton, etc.). The strategy of collecting field data through interviews with producers in the region is necessary because, in the case of its sequential crops, a visit made on a single date does not identify the complete production system. In most studies, however, this strategy is limited because it does not allow collecting a large number of samples over large areas and remains problematic for poorly represented classes, such as integrated systems.

In this study, cropping practices samples for the 2012–2019 period were collected from five different sources: in situ GPS field limits and surveys, interviews with producers, data from local cooperatives, consultants and from research projects. During interviews with producers of the URT (Technological Reference Units in partnership with Embrapa Agrossilvipastoril), data on field limits and on the history of the cropping practices could be collected. Samples were also gathered from the *Bom Futuro* group, one of the largest agricultural production groups in Brazil, and from other research initiatives [56]. The final database totaled 1748 polygons spread over 46 state municipalities. Table 1 indicates the percentage of samples (by hierarchical level) that each data source represents.

**Table 1.** Source and number of field samples (training/validation database) for all classes and levels.

| | Level 1 | | | Level 2 | | Level 3 | |
|---|---|---|---|---|---|---|---|
| **Source** | **SC** [1] | **DC** [2] | **Sce** [3] | **iCL** [4] | **Sco** [5] | **iCL1** [6] | **iCL2** [7] |
| In-situ GPS | 10.81% | 12.81% | 2.62% | 14.66% | 0.09% | 26.46% | 0.79% |
| Bom Futuro Group | 9.04% | 35.93% | 11.43% | 12.22% | 25.57% | 22.05% | 8.98% |
| Embrapa | 0.17% | 6.64% | 3.32% | 5.5% | 0.00% | 9.92% | 1.42% |
| Consultants | 2.12% | 13.73% | 4.1% | 7.16% | 0.00% | 12.91% | 17.48% |
| Previous Study [8] | 0% | 8.75% | 8.9% | 0% | 4.45% | 0% | 0% |
| **TOTAL** | **22.14%** | **77.86%** | **30.37%** | **39.53%** | **30.10%** | **71.34%** | **28.66%** |

[1] Single Crop (SC); [2] Double Crop (DC); [3] Soybean+Cereal (SCe; mainly corn but also millet, sorghum or sunflower); [4] Soybean+Pasture (iCL); [5] Soybean+Cotton (SCo); [6] Soybean+Pasture (iCL1); [7] Soybean+Pasture in association with cereals (iCL2); [8] Data obtained on previous study [56].

In order to avoid the border effect, all the polygons were overlaid on MODIS images, while the border pixels were removed by applying a −250 m buffer. The spatial and temporal distribution of the field samples is given in Figure 3 for Single Crop (SC); Double Crop (DC); Soybean+Cereal (SCe; mainly corn but also millet, sorghum or sunflower); Soybean+Pasture (iCL); Soybean+Cotton (SCo); Soybean+Pasture (iCL1); Soybean+Pasture in association with cereals (iCL2).

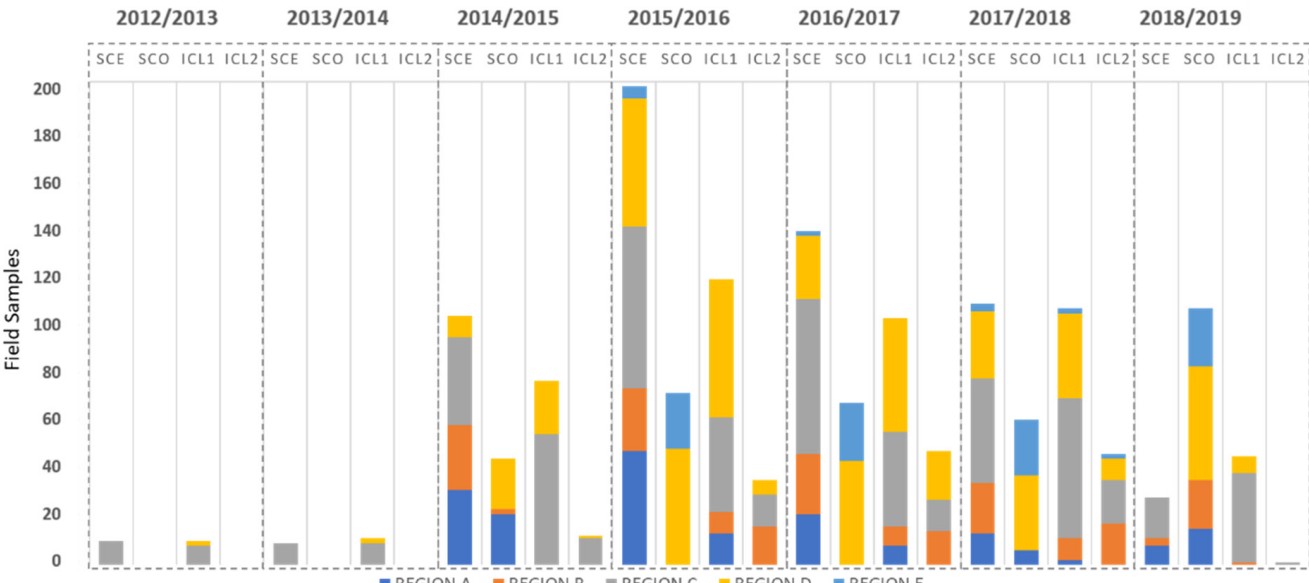

**Figure 3.** Spatio-temporal distribution of field samples by the four main classes according to each crop year and agro-climatic regions.

The training dataset was composed of 2/3 of the fields, randomly selected; the remaining 1/3 of the fields were used to validate the classification map. The separation between the training and validation datasets was performed at the field level, not at the pixel level, to avoid classification bias (due to training and validation samples located in the same field).

## 4. Method

### 4.1. The Classification Strategy

Figure 4 shows the flowchart of the classification strategy adopted in this paper. The classification is based on the MODIS SITS acquired for the mid-2012–mid-2019 period and the field samples database, a hierarchical structure and Random Forest algorithm (RF). The strategy consists of testing different sample-balancing strategies and assessing the impact of the spatial and temporal distributions of the samples on the classification accuracy of the cropping systems. The classification strategy with the highest accuracy for the iCL classes is then used to produce annual cropping systems maps at the Mato Grosso scale for the whole period.

### 4.2. The Hierarchical Image Classification

We adopted a hierarchical image classification approach [57] at four levels (Figure 5):

Level 0: The starting map corresponds to the Soybean class of the MapBiomas collection product [39];

Level 1: The Soybean class is separated into two classes: Single Crop (SC) and Double Crop (DC);

Level 2: The Double Crop class is separated into three classes: Soybean+Cotton (SCo), Soybean+Cereal (SCe; mainly corn but also millet, sorghum or sunflower), and Soybean+Pasture (iCL);

Level 3: The integrated Soybean+Pasture class is separated into two classes: Soybean+Pasture (iCL1) and Soybean+Pasture in association with cereals (iCL2).

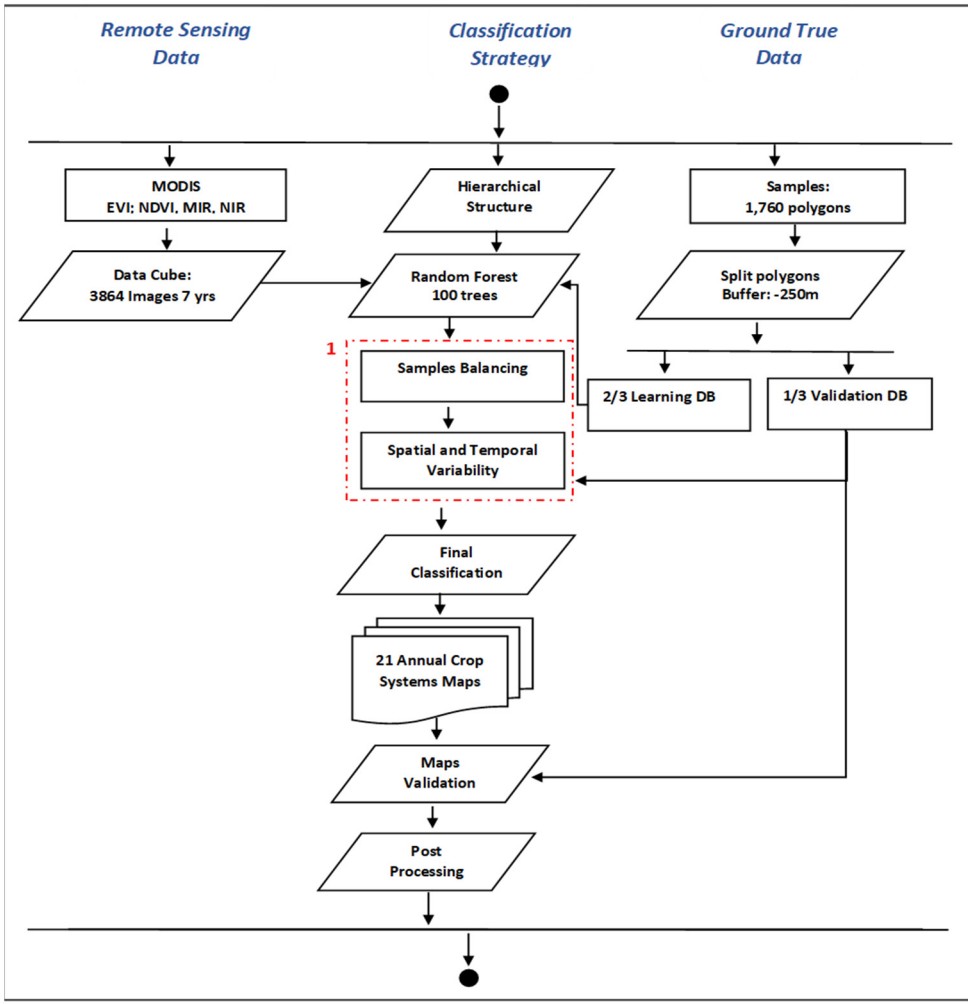

**Figure 4.** Flowchart presenting the main classification strategy issues. [1] Methodological issues: Samples balancing strategy assessment; Spatial variability assessment; Temporal variability assessment.

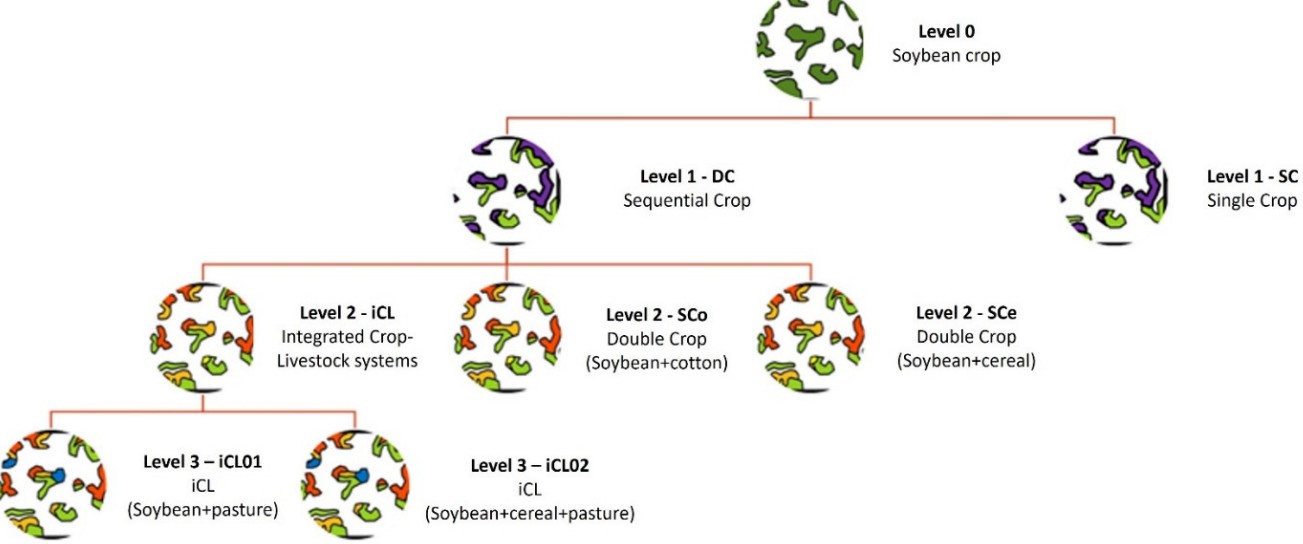

**Figure 5.** Hierarchical classification: the four classification levels and the associated cropping systems classes.

The classification algorithm was based on a machine learning technique performed in cloud computing for annual mapping, where each map represents the cropping system of an entire crop year. Based on the classification experiments carried out by Kuchler et al. [31], who tested two classification algorithms and different spectral features for classifying Mato Grosso cropping systems, we opted for an RF classifier [58] with 100 random trees. The classification process was based on four features, two spectral indexes and two spectral bands (the Normalized Difference Vegetation Index (NDVI), Enhanced Vegetation Index (EVI), Near-Infrared spectral band (NIR) and Mid-Infrared spectral band (MIR)) of the MOD13Q1 product and on the learning/validation dataset of approximately 25,000 pixels. The platform used to run the RF classification was the Google Earth Engine, and we were thus able to easily process the 3864 images of the dataset for the whole Mato Grosso state and for the mid-2012–mid-2019 period (23 images per year; 4 bands; 6 tiles; 7 years).

### 4.3. The Training Databases

#### 4.3.1. Balanced and Unbalanced Samples

The sample balancing strategy is fundamental as it directly influences the result of the RF classification, particularly for the "rare" classes. Unbalanced data refers to a situation where the number of observations is not the same for all classes. Machine learning classifiers aimed at minimizing the overall error rate tend to favor the class with the highest proportion of observations (or majority class). This can be particularly problematic for classifying a "rare" class (or minority class), such as the iCL class. For this reason, it is essential to perform a sensitivity analysis of the classifier performance to the class distribution of the training samples.

Based on the sample balancing practices and recommendations found in the literature for RF classification, three scenarios were tested: (i) balanced dataset (*Bset01*), using a similar number of samples in each class, Dalponte et al. [59] and Jin et al. [60], (ii) unbalanced dataset (*Bset02*), using a distribution of samples close to the actual distribution of classes found in the field as estimated from the collected dataset, Colditz et al. [61] and Mellor et al. [62], and (iii) unbalanced dataset (*Bset03*), using an over-representation of the "rare" class iCL. Figure 6 shows the distribution of samples for each scenario and the results obtained for the iCL class.

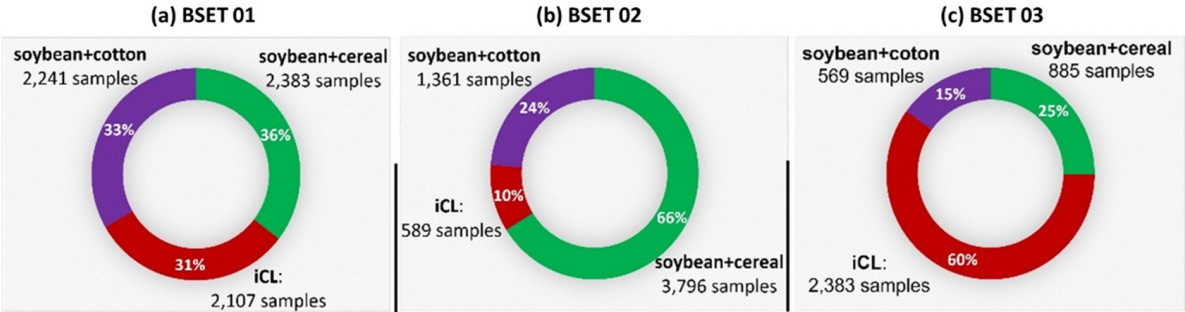

**Figure 6.** Composition of the training samples data-base (in % of the number of pixels), according to the three scenarios: (**a**) Bset01: Balanced number of samples per class; (**b**) Bset02: Unbalanced samples with a class distribution close to the one found in the field; (**c**) Bset03: Unbalanced samples with an over-representation of the iCL class.

#### 4.3.2. Spatio-Temporal Distribution of the Samples

We showed previously that the training/validation datasets were distributed very unevenly in space and time. Testing the performance and robustness of classifications trained with samples acquired at different years and in different regions was a mandatory pre-classification step, in particular for Mato Grosso, which displays a great diversity of environments and agriculture production systems.

To study the impact of this spatio-temporal distribution of the datasets on the accuracy and robustness of the classifications, we tested and evaluated the accuracy of the cropping system maps according to different years, characterized mainly by different rainfall patterns (Figure 2) and agro-climatic regions of Mato Grosso (Figure 1).

To assess the impact of the uneven temporal distribution of the samples datasets, we compared the classification accuracy obtained using two learning database construction strategies:

1.  Strategy (a): Two-thirds of the field samples acquired for a given year are used as training data to classify SITS for the same year, the other third is for validation;
2.  Strategy (b): Two-thirds of the samples collected during the different years (between 2014/15 and 2018/19) are pooled and then used to train the SITS for each of the years between 2012/13 and 2018/19. For 2014/15 to 2018/19, the validation is done with the remaining samples collected for the year in question (samples not used for training), while for the 2012/13 and 2013/14 years, the validation is done with the samples acquired during those years, because there were not enough samples for those years.
3.  To assess the impact of the uneven spatial distribution of the samples datasets, we processed them the same way:
4.  Strategy (a): Two-thirds of the field samples acquired for a given agroclimatic region were used as training data for the same given region, the other third was used for validation; Zone E was not concerned by this strategy because of the low number of samples in this zone;
5.  Strategy (b): Two-thirds of the samples collected in all regions are pooled and then used to train the SITS for each region. The validation is done with the remaining samples collected for the region in question (samples not used for training).

### *4.4. Post Processing*

Annual maps of farming systems were produced at the pixel scale. A spatial filter was then applied to the iCL class in order to eliminate the incorrectly classified isolated pixels or groups of pixels (essentially mixed pixels related to the transitions between the crop, pasture and forest classes) [63,64]. This spatial filter excludes pixels where a minimum of five other pixels of the same class are not connected, i.e., patches with an area smaller than approximately 35 ha. This number, which may appear high, should be evaluated in regard to the average (90 ha) and minimum (30 ha) size of the fields in the dataset.

### 5. Results

### *5.1. Impact of Balanced and Unbalanced Learning Database*

Table 2 presents the classification accuracy performances, obtained for Level 02, level with most classes, including iCL, using the three strategies of the training database: Bset1 (balanced dataset), Bset2 and Bset3 (unbalanced datasets), for the classes Soybean+Cotton, Soybean+Cereals, and the integrated crop–livestock system.

**Table 2.** Class accuracy metrics at Level 02, obtained with balanced (Bset1) and unbalanced (Bset2 and Bset3) training datasets. The classes are the Soybean+Cotton (SCo), Soybean+Cereals (SCe) and integrated systems (iCL).

| | Bset 1 | | | Bset 2 | | | Bset 3 | | |
|---|---|---|---|---|---|---|---|---|---|
| **Metrics** | **SCo** | **SCe** | **iCL** | **SCo** | **SCe** | **iCL** | **SCo** | **SCe** | **iCL** |
| Producer accuracy | 1 | 0.81 | 0.8 | 1 | 0.91 | 0.76 | 0.99 | 0.54 | 0.88 |
| User accuracy | 1 | 0.8 | 0.81 | 1 | 0.79 | 0.9 | 1 | 0.82 | 0.66 |
| F-Score | 1 | 0.96 | 0.81 | 1 | 0.85 | 0.83 | 1 | 0.65 | 0.75 |
| Nber of samples (pixels) | 2241 | 2383 | 2107 | 1361 | 3796 | 589 | 2383 | 885 | 2383 |
| **Overall accuracy** | | **0.87** | | | **0.89** | | | **0.81** | |

The results show that the double cropping system Soybean+Cotton (SCo) is classified with a very high accuracy, whatever the training strategy, while the F-Score of the Soybean+Cereals (SCe) class is between 0.65 for Bset3 and 0.96 for Bset1. The best F-scores for the iCL class were obtained for Bset01 and Bset02 (0.81 and 0.83, respectively), but with a higher commission error for Bset01 (+9%).

The effectiveness of the Bset2 sampling strategy is probably related to the intrinsic variability of the majority classes with respect to the characteristics of the Random Forest classifier, which can suitably deal with such variability (i.e., building multiple different trees for the same class) at the need of a larger sample set for those classes. Better discrimination of such classes (SCe in our case) ends up in improving performances also on the minority iCL class.

Due to higher overall precision, higher F-score and lower commission error for the iCL class, the Bset2 unbalanced training dataset, with a number of samples per class representative of the real proportion of classes in the field, is used for the following classifications at Levels 02 and 03.

### *5.2. Impact of Temporal Distribution of the Learning Database*

#### 5.2.1. The NDVI Inter-Annual Variability: Example of the iCL Class

Before quantifying the impact of the uneven temporal distribution of the field samples on the accuracy of the multi-year classifications, we represented the interannual variability of the NDVI of the iCL class samples. The mean annual NDVI time series displays a strong seasonality (Figure 7), with two vegetation peaks corresponding to the wet season (from October to March; period of soybean cultivation) and to the dry season (from April to September; pasture period). The NDVI inter-annual variability is more pronounced during the wet season, in particular for years of high ENSO intensity, with two peaks of NDVI growth in 2015/2016 (related to the heterogeneity of rainfall and/or the quality of images), and an extended plateau in 2018/2019. NDVI profiles are more homogeneous during the dry season from June to August, even in an ENSO year.

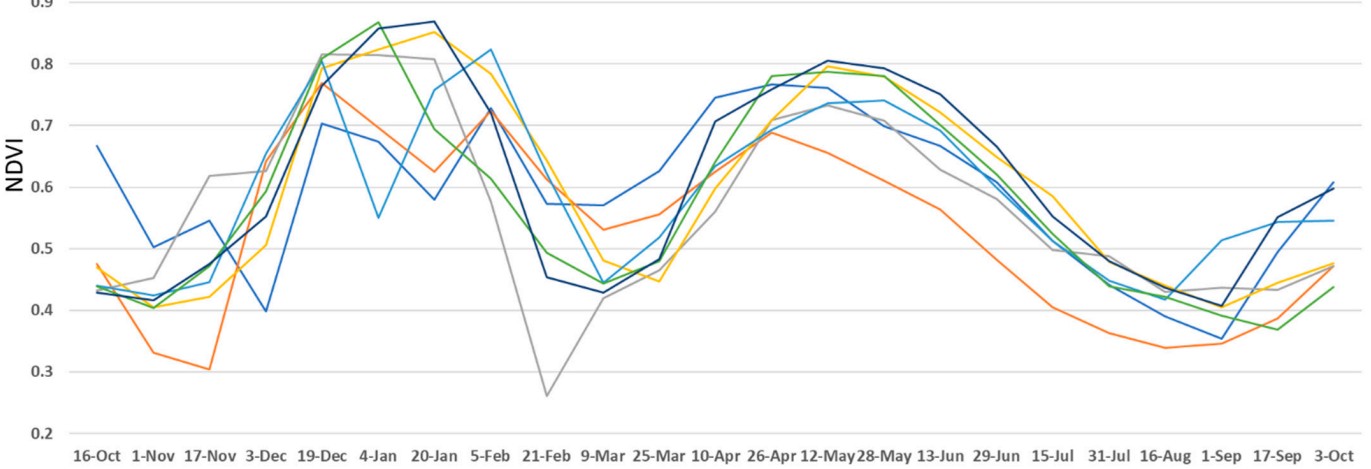

**Figure 7.** Inter-annual variability of the mean NDVI time series of the iCL class samples over Mato Grosso, for seven agricultural campaigns: 2012/2013, 2013/2014, 2014/15, 2015/16 (ENSO), 2016/17, 2017/18 and 2018/19 (ENSO).

#### 5.2.2. Impact of the Temporal Training Strategy on the Classification Performances

In Figure 8, the results show that the overall accuracy and the F-score of the iCL class are higher for strategy (b), with pooled annual samples than for strategy (a), with individual annual samples. This is particularly true for years with small sample datasets, such as 2012/2013, 2013/2014 and 2018/2019. For years with large samples datasets (2015/2016, 2016/2017 and 2017/2018), the overall precision and the F-score have little

variation; regardless of the number of years used for training. The results suggest that the classification accuracy is more related to the number of training samples than to the inter-annual variability of the precipitation.

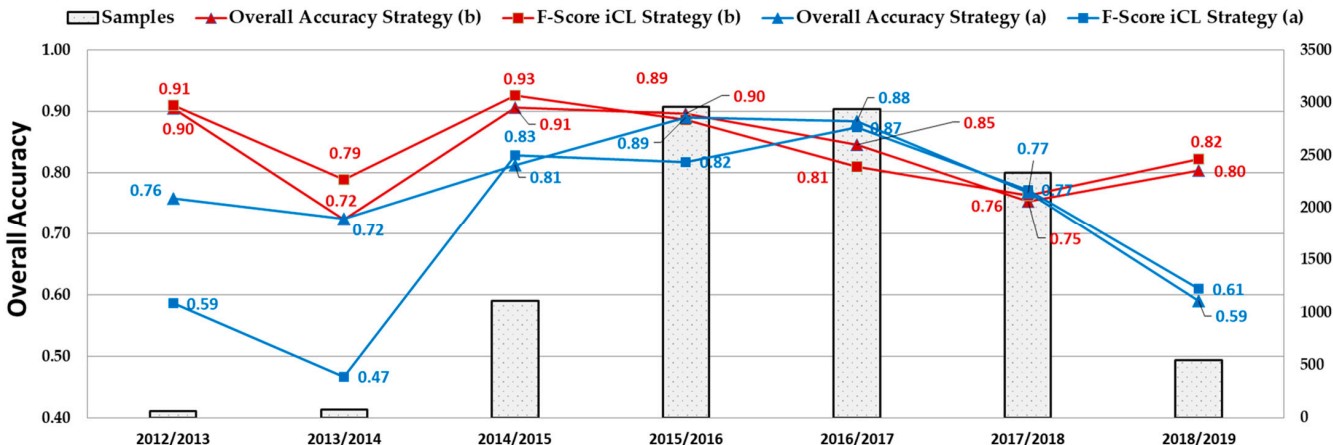

**Figure 8.** Overall accuracy and F-score of the iCL class obtained for each year and for the two temporal sampling strategies is shown in the top part; the number of samples (given in pixels) available per year is represented by the histograms in the lower part of the figure.

### 5.3. Impact of Spatial Distribution of the Learning Database

#### 5.3.1. The NDVI Inter-Regional Variability: Example of the iCL Class

The NDVI annual profile of the iCL class displays differences between agro-climatic regions (Figure 9), especially during the rainy season (Dec–Feb). Region E has the "noisiest" NDVI curve, with the greatest amplitude, but it is also the zones that contain the fewest samples. During the dry season (June–August), zones C and D behave similarly, despite the large difference in the number of samples. The dispersion and symmetry of the NDVI distribution by region do not show a clear trend. In zone A, the amplitude is greater during the rainy season, while for zones B, C and E, a greater diversity of values was found in the dry season. In zone D, the amplitude does not vary significantly. Finally, all the regional curves show that the average NDVI values between the rainy season and the dry season are not significantly different (with slightly lower values in the dry season for regions B and C).

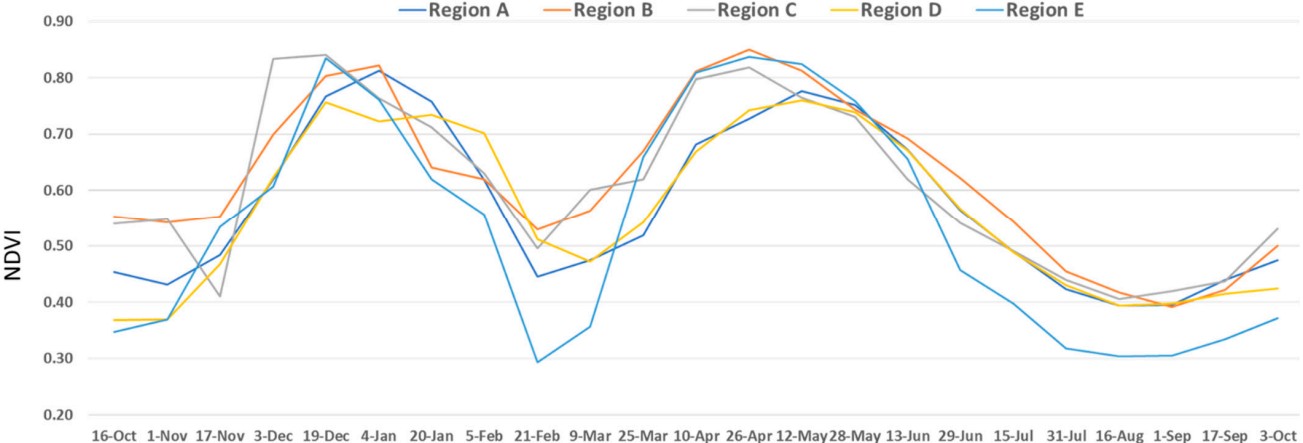

**Figure 9.** Spatial variability of the mean annual NDVI time series (between mid-2012 and mid-2019) of the iCL class samples over the five agro-climatic regions of Mato Grosso produced in previous study [36].

### 5.3.2. Impact of the Spatial Training Strategy on the Classification Performances

Analysis Figure 10 allows the assessment of the impact of the spatial sampling strategy on the overall accuracy of the map and on the F-score of the iCL class. We observed that the larger the number of samples in the region is, the higher the overall accuracy and the F-score, both for training strategy (a), with regional samples, and for training strategy (b), with global pooled samples. The overall accuracy and F-score values in zones B, C and D are close for both strategies. In zone A, when all years are used for training (dataset b), there is a significant improvement in the overall accuracy. Following the pattern of the previous test, the results suggest that the classification accuracy is more related to the number of samples for training than to the climatic variability present in the different regions of the state.

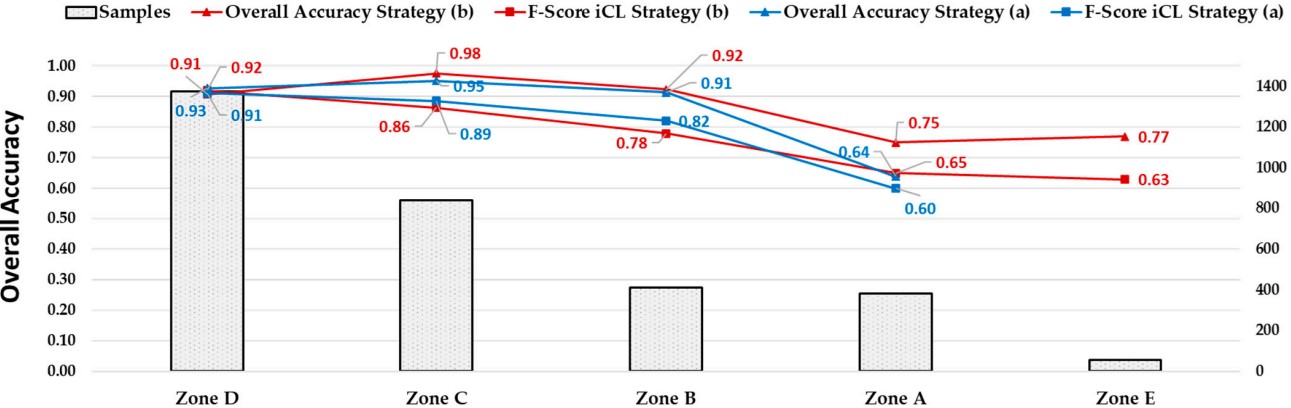

**Figure 10.** Global Accuracy (OA) and F-score of the iCL class, obtained for the five climatic regions (A to E) and for the two spatial sampling strategies (datasets a and b). The histograms show the number of samples (given in pixels) available per region.

### 5.4. Assessment of Cropping System Classifications at the Different Levels

#### 5.4.1. Global Accuracy

Given the previous results, the annual classification of the cropping systems was trained with the unbalanced dataset (Bset2), and with all the available samples (aggregated for all regions and years). The classification accuracy was assessed at each level of the hierarchy.

The overall accuracy of the classifications of cropping systems obtained at the three levels (the score was limited in the classes of each hierarchical level) is between 0.70 and 0.96 (Table 3). As expected, Level 01 (DC and SC classes) obtains, on average, the best score (0.89) and shows the greatest multi-year stability (between 0.86 and 0.96). Next comes Level 02 (0.84 on average), then Level 03 (0.77 on average). Levels 02 and 03 have amplitudes of variation within the same order.

**Table 3.** Overall accuracy of the annual hierarchical classification of the cropping systems, at the three levels, for 7 years in Mato Grosso.

| | Overall Accuracy | | |
|---|---|---|---|
| **Agricultural Year** | **Level 01** | **Level 02** | **Level 03** |
| 2012/2013 | 0.87 | 0.91 | 0.72 |
| 2013/2014 | 0.86 | 0.72 | 0.70 |
| 2014/2015 | 0.89 | 0.91 | 0.73 |
| 2015/2016 | 0.87 | 0.89 | 0.74 |
| 2016/2017 | 0.89 | 0.85 | 0.92 |
| 2017/2018 | 0.90 | 0.77 | 0.80 |
| 2018/2019 | 0.96 | 0.82 | 0.79 |
| **7-year mean** | **0.89** | **0.84** | **0.77** |
| **7-year standard deviation** | **0.03** | **0.07** | **0.07** |

### 5.4.2. Integrated System Classes Accuracy

The F-scores of the double crop and iCL classes calculated for each year are given in Figure 11. The DC, iCL and iCL1 classes, although belonging to different levels, have similar accuracy (respectively, 0.89, 0.85 and 0.84). The iCL2 class is less well ranked (0.63).

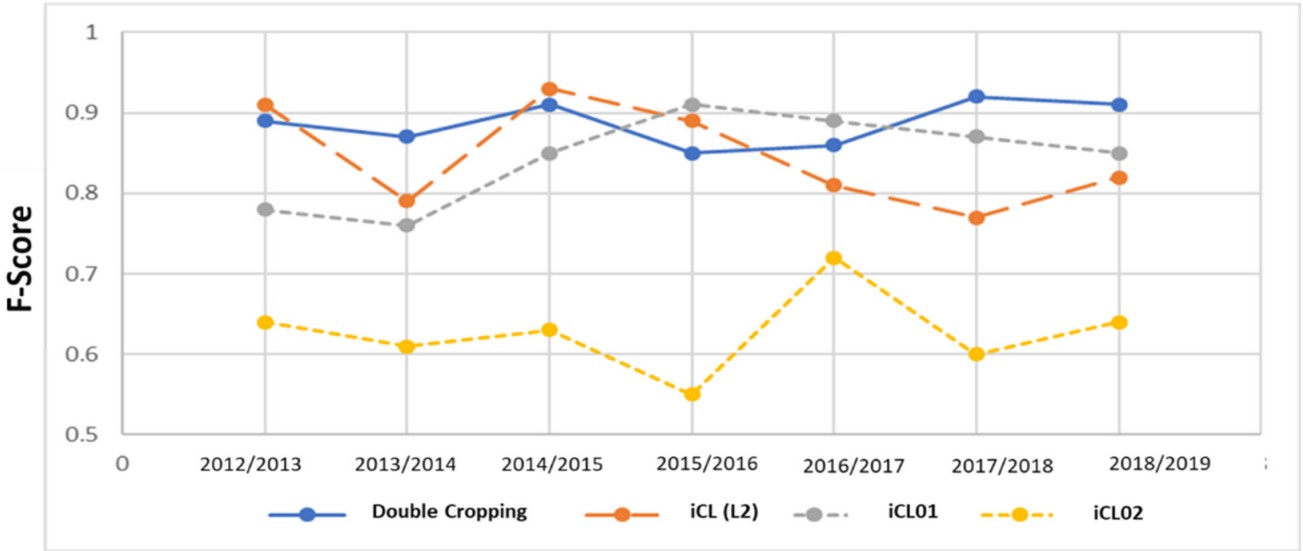

**Figure 11.** Annual F-score of the double crop and iCL classes of Level 01 (blue solid line), Level 02 (orange large dotted line) and Level 03 (gray and yellow dotted line), calculated at the Mato Grosso scale between 2012–2013 and 2018–2019.

At Level 02, the iCL class has a user accuracy between 0.8 and 0.94 (and. therefore, a low commission error) and a producer precision between 0.68 and 0.88 (and, therefore, a mean error of omission). The large number of false negatives suggest that the area implanted with the iCLs is underestimated (some iCL areas have been classified as Soybean+Cereals); on the other hand, the low number of false positives (of the Soybean+Cereals areas classified as iCL) indicates more reliability when a pixel is classified as iCL. This good result is partly due to the sample balancing strategy presented above (choice of a distribution close to that found in reality).

The difference in classification accuracy between iCL1 and iCL2 can be explained by the lack of available samples for the iCL2 class in addition to these samples being more concentrated in time and space, with less spectro-temporal diversity, as can be seen in Figure 4.

### 5.5. The Mato Grosso Cropping System Maps

#### 5.5.1. The Final Classification Products

Based on the methodology, 21 maps were derived (7 years and 3 levels); Figure 12 shows the 2018–2019 cropping system map of Mato Grosso at Level 02, as an example of the methodology output. Those maps can be used to analyze the land use dynamic and iCL distribution over the years in different regions.

#### 5.5.2. Evaluation of the Cropping System Maps with Official Statistics

The official harvest statistics (IBGE-PAM) [40], based on annual surveys carried out at the municipal level, are only available for cotton and corn, sunflower and/or sorghum crops. It is why we could only evaluate the classes Soybean+Cotton, and Soybean+Cereals (Figure 13).

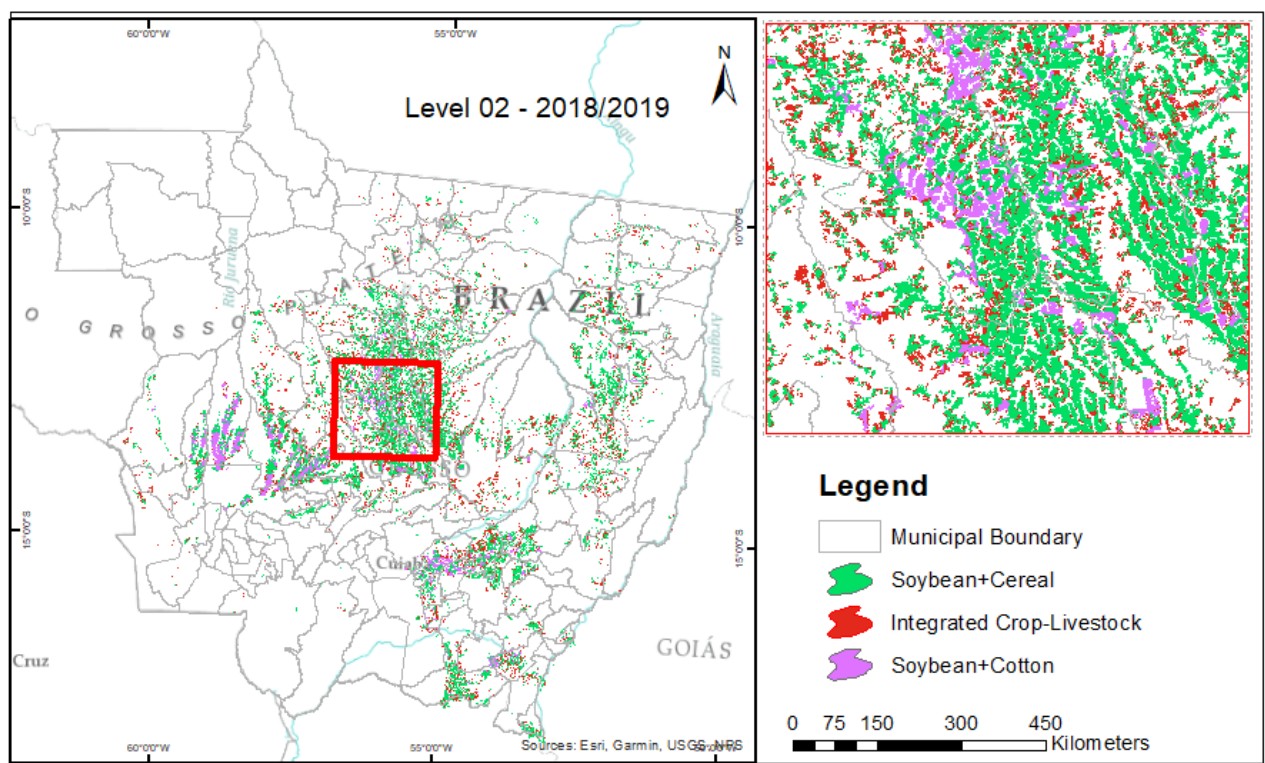

**Figure 12.** 2018/2019 Level 2 classification map at the Mato Grosso scale and a detailed area.

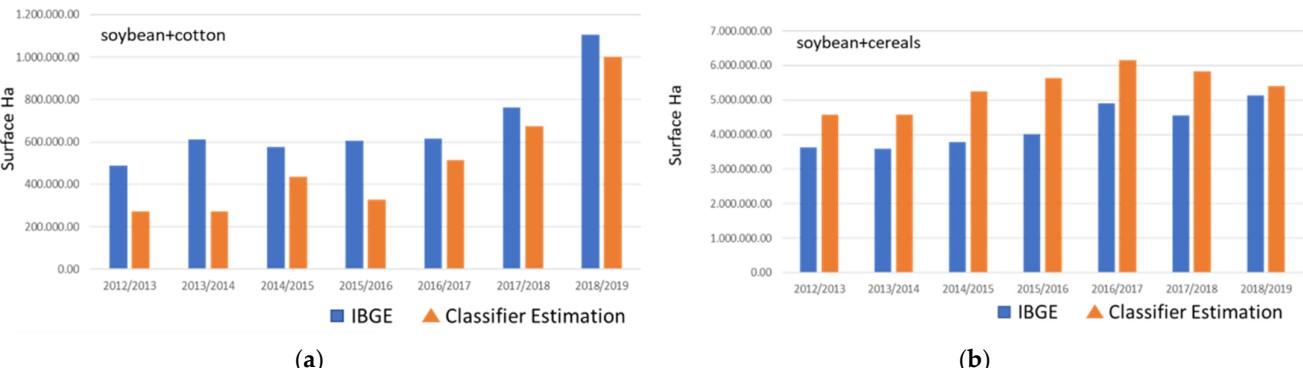

(**a**)　　　　　　　　　　　　　　　　　　　(**b**)

**Figure 13.** Comparisons between the annual cultivated areas of Mato Grosso given by IBGE-PAM and those obtained by Random Forest classification, for (**a**) cotton (class SCo), and (**b**) cereals (class SCe).

At the state scale, the correlation between the annual cotton areas obtained by IBGE-PAM and those obtained by image classification is 0.94. For the SCe class, the correlation is 0.74, with an overestimation of the area provided by the classification. This overestimation is due to the inclusion of millet as class SCe (Soybean+Cereal) in our classification, while the millet class is not available in the IBGE database. In addition, the areas provided by IBGE-PAM are calculated based on interviews.

Regarding the class iCL, which is the target of our work, IBGE does not have the means to provide that information, but other reference values can be found at "*Rede ILPF*". *Rede ILPF* is an iCLF network in Brazil created by public and private companies. They were initiated in 2012 and aimed to accelerate the widespread adoption of integrated crop–livestock–forest (iCLF) technologies by rural producers. They organized a huge survey interviewing large farmers by telephone. This survey estimated for Mato Grosso an iCL area of 1.25 Mha in the 2015/2016 crop year; this value is similar to the 1.37 Mha obtained in our study based on remote sensing techniques.

## 6. General Discussion

*6.1. The Integrated Crop–Livestock System: A Remote-Sensing Challenge*

To meet the public policy needs related to Low Carbon Agriculture (Plan ABC) of the Brazilian state, in particular the monitoring of the integrated agriculture–livestock systems (iCL), we have developed a method to produce annual maps of cropping systems at larges areas. To do this, we had to overcome methodological and technical challenges: Mapping a complex and rare land-use class, retrospectively and at the regional scale, and on an operational mode.

### 6.1.1. Mapping Rare Complex Agricultural Systems

Integrated agricultural production systems cover different productive strategies with various spatio-temporal arrangements of agricultural, pastoral and forestry activities (sequential, associated crops, rotation, etc.). These are complex systems, difficult to map by remote sensing. In this study, understanding the spatial and temporal functioning of the integrated systems gave us the keys to move from land cover, as observed by satellite, to land use, which is the marker of socio-economic activity.

As a rare class, there is a need to work on the land use training data. We analyzed three different compositions of training data and found that the unbalanced dataset, with a similar composition to reality, gave the best classification results for the iCL class. This result is in accordance with Colditz et al. [61] and Mellor et al. [62], who showed better Random Forest classification performance when the distribution of samples between classes is closer to the actual proportion of the classes in the field. However, it is in contradiction with other studies [60,65] that showed better classification results with balanced learning samples. These opposing bibliographic results must be related to the richness of the training samples and even in relation to the distribution and proportionality of minority classes.

### 6.1.2. Classify Agricultural Systems Retrospectively at Regional Scale

Testing the performance and robustness of classifications trained with samples acquired at different years and in different regions is an essential pre-classification step, in particular for areas with a great diversity of environments and agriculture production systems, such as Mato Grosso state. For the temporal (and spatial) sensitivity of the training dataset, we tested two scenarios of classification, one using the associated annual (agro-climatic regional) dataset to classify each year (each region), the other using the whole training dataset to classify each year (each agroclimatic region). The results showed that, in both cases, the classification accuracy was more related to the number of training samples than to the inter-annual or regional variability of the environmental conditions, suggesting the pooling of the samples for the classification. This training strategy finding is in accordance with Picolli et al. [27], who obtained a good classification accuracy for double cropping, using a unique learning and validation base of 1800 points for 15 years and for different regions of the state of Mato Grosso. This result might seem to be in contradiction with the results of Parente et al. [54,55], who showed the importance of considering the spatial and temporal spectral variability of the pasture class in Brazil, but in fact, it is not, as Parent et al. [54,55] artificially increased the number of samples by photo-interpretation for each year of study and each scene of the national territory. This could not be done in our study, as photo-interpreting the iCL class is not possible.

### 6.1.3. Processing a Large Volume of Data

Over the period studied, a total of 3864 images were processed. The image processing was facilitated by the use of the Google Earth Engine (GEE) platform, which offers easy access to images and which makes it possible to create processing chains that are easily reusable on other datasets (regions or years). The solution proposed in this work is based on Big Data technologies—Cloud computing and the use of machine learning algorithms applied to Earth observation data. This approach has proven to have great potential to

respond to the challenges previously posed in the field of remote sensing and allows the development of methods for operational purposes.

### 6.2. Perspectives to Map Low Carbon Agriculture in Brazil

#### 6.2.1. Research Perspectives

The dynamic of the Earth observation and science data tools opens up many perspectives for methodological research. Thus, we can consider improving the image processing chain by including other data sources that have a better spatial resolution (e.g., Landsat, CBERS, Sentinel-2) or that are less sensitive to cloud cover, such as radar images (e.g., Sentinel-1). The data from the European Sentinel satellites of the COPERNICUS program are extremely important in terms of mapping agricultural practices by their high spatial and temporal resolution and by their spectral richness. The first Sentinel-2 satellite was launched in 2015, but the 5-day revisit frequency was not reached until March 2018 for Brazil, limiting the use of these satellites for retrospective analysis. Thus, the advantage of MODIS remains the consistency of the dataset over time since 2000, with a high temporal frequency.

There are different multi-sensor data fusion pathways that could be tested for mapping iCLF and to better distinguish the different iCLs systems. New machine learning algorithms make it possible to process heterogeneous, multi-source data. CNN (Convolutional Neural Networks)-type algorithms eliminate the need for the choice and calculation of descriptors, and LSTM (Long Short-Term Memory)-type algorithms allow better consideration of time information. However, the extraction power of these new tools is conditioned by learning the underlying models, which requires a large amount of quality data. To date, the localized datasets available on agricultural systems are not yet sufficient to take advantage of this computing power.

#### 6.2.2. Operational Perspectives

The possibility of using a large volume of Earth observation data with high-performance computers and parallel processing has become popular, and its investment costs have considerably decreased [33,66,67]. This drives the creation of structures based on Big Earth Observation Data solutions. The combination of the Big Data and Earth Observation areas enhances the use of time series of satellite images with the creation of data cube structures, a multidimensional array of values (space, time, property) with great dimensionality [68], enabling the identification of hidden patterns in large amounts of data to be transformed into information [69].

Finally, the use of the Google Earth Engine allowed the development of the methodology, with all the positive aspects already mentioned; however, this use poses the problem of a solution proposed by a private actor who can modify their data and platform access policy, and which stores the data and products. This type of risk is less in collaborative structures, such as the Orfeo ToolBox (OTB) in France, or in national structures, such as the structure under development of the Brazilian Data Cube, which will be the preferred option for the pursuit of this work.

### 7. Conclusions

In this work, we developed a satellite image processing chain based on the classification of MODIS time series for annual mapping of the integrated crop–livestock systems (iCL) at a regional scale for seven agricultural seasons (2012–2019). The Random Forest classification strategy was designed and tested in six agro-climatic regions of Mato Grosso state, the biggest soybean producer in Brazil. Ground data with a heterogeneous distribution in space and time was acquired in order to assess the spatio-temporal variability impact on the classification results. The results show that the MODIS time series combined with machine learning algorithms are capable of detecting sequential crops, especially integrated crop–livestock systems, and that parallel computing in the cloud, as part of the Big Earth Observation Data concept, allows the development of an efficient and reproducible

methodology for mapping complex systems over large areas. This methodology can be applied as an operational tool for monitoring the adoption of sustainable intensification practices recognized by Brazil´s Agriculture Low Carbon Plan (ABC Plan).

**Author Contributions:** Conceptualization, P.C.K., M.S., A.B., R.F., D.A. and P.L.O.d.A.M.; methodology, P.C.K., M.S., A.B., R.F., D.A. and R.G.; software, P.C.K., M.S., A.B. and M.R., supervision, M.S. and A.B. All authors have read and agreed to the published version of the manuscript.

**Funding:** The main author received a scholar fellowship from the Capes (Coordenação de Aperfeiçoamento de Pessoal de Nível)-Cofecub (Coopération Universitaire et Scientifique avec le Brésil) GeoABC Project (Methodologies and technological innovation for satellite monitoring of low carbon agriculture in support of Brazil's ABC Plan, project No. 845/15). The ground visits were partly supported by the H2020- MSCA-RISE-2015 ODYSSEA European project (Project Reference: 691053) and the French Agricultural Research Centre for International Development (CIRAD).

**Institutional Review Board Statement:** Not applicable.

**Informed Consent Statement:** Not applicable.

**Data Availability Statement:** Not applicable.

**Acknowledgments:** We thank the Capes-Cofecub, CIRAD-TETIS Unit, for the welcoming stay in Montpellier and methodological support, and the teams from UERJ/PPGMA, EMBRAPA Solos, EMBRAPA Agrosilvipastoril, Embrapa Labex Europe, as well as the IEA CNRS SCOLTEL project for their expertize and help. We also thank the Bom Futuro Group and the consulting Adriano da Paz for providing field data.

**Conflicts of Interest:** The authors declare no conflict of interest.

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
