# Peer review of "Monitoring Complex Integrated Crop–Livestock Systems at Regional Scale in Brazil: A Big Earth Observation Data Approach"

_remotesensing, doi:10.3390/rs14071648_

Round 1

Reviewer 1 Report

The undertaken work is very interesting and can be considered novel and will help us understand the utility of EO data in quantifying natural phenomena. I found this paper very interesting where several technical aspects were nicely implemented and explained sufficiently. Undoubtedly, authors invested huge amount of time and have made a great effort to produce this high-quality of research which is clearly structured and the language used is largely appropriate. As final decision, I see that this manuscript in its form and level deserves to be accepted for publication in MDPI-RS BUT after addressing below MINOR COMMENTS.

COMMENTS:

  • I suggest summarising a bit the title of the paper would be great and rend the paper more interesting to read.
  • I found the abstract very well written and structured.
  • Also the introduction is well structured and organised.
  • In fig.1 try to adopt the same size for both photos.
  • Extend the title of fig2.
  • 4 the flowchart is not clear, please deliver high quality figures, and the text in the flowchart must be added inside a square or a shape, your flowchart is very good just make it more organised.
  • Please I insisted on the quality of the figures 4, 5 and others.
  • Please a general recommendation and perspective to your conclusion as you have done in the abstract.
  • As final general comment, please make sure to define ALL the acronyms form their first appearance in the paper. Also, all the references MUST BE CHECKED and formatted as required by MDPI- RS, also make sure that all the references have DOI number unless it is not available.

Author Response

Dear Sir or Madam,

Thank you for your careful reading, as well as the proposition of very important points for improving our work. Please find bellow a point-by-point response.

  • I suggest summarising a bit the title of the paper would be great and rend the paper more interesting to read.

R.: Thank you very much for the suggestion. We reworked the title: “Monitoring complexes integrated crop-livestock systems at regional scale in Brazil: A big earth observation data approach”

**as we changed the title as suggested, we are thinking removing from Keywords (Line 35,36): Crop Livestock integration and iCL (As it is already mentioned in the new title) and adds training sample designs and samples balancing (As those are related to the developed methodology).  

  • I found the abstract very well written and structured.
  • Also the introduction is well structured and organised.

R.: Thank you very much, your feedback is very important.

  • In fig.1 try to adopt the same size for both photos.

R.: Figure 1 has been slightly modified (the resolution was increased in resolution, and  maps were resized). Thank you.

  • Extend the title of fig2.

R.: Thanks for the sugestion. We rewrote the title of the figure 2, which is now presented as follows: “Histogram depicting monthly rainfall distribution for Mato Grosso at different years

  • 4 the flowchart is not clear, please deliver high quality figures, and the text in the flowchart must be added inside a square or a shape, your flowchart is very good just make it more organised.

R.: Done. We enlarged the font and reworded the text facilitating the understanding.

  • Please I insisted on the quality of the figures 4, 5 and others.

R.: Really we had to rework the figures and significantly increase the resolution. Thanks for the suggestion.

  • Please a general recommendation and perspective to your conclusion as you have done in the abstract.

R.: Thank you for your observation, we totally agree with you. We changed the conclusion in order to sumarize the methology and also to include a perspective for monitoring the integrated systems as we have being discusing in Brazil.

As final general comment, please make sure to define ALL the acronyms form their first appearance in the paper. Also, all the references MUST BE CHECKED and formatted as required by MDPI- RS, also make sure that all the references have DOI number unless it is not available.

R.: Thanks for the observation. We tweaked the acronyms, there really were some that needed tweaking. Also for the bibliography. Now, all articles cited are with DOI. References that do not present DOI refer to official data and references collected in public systems and projects. Formatting is following the pattern indicated by MDPI-RS, and it was implemented with the help of the Zotero tool.

Thank you so much.

Reviewer 2 Report

The overall objective of this study was to develop a classification strategy for annual mapping of integrated Crop-Livestock systems (iCL) at regional scale. This strategy was designed and tested in the six agro-climatic regions of Mato Grosso, using MODIS satellite images time series acquired between 2012 and 2019, ground data with heterogeneous distribution in space and time, and a Random Forest classifier. The experimental process is rigorous and fully demonstrated. But I think there are still the following problems:

  1. The resolution of Figure 1 is too low to give the reader a good look and feel.
  2. The font in the flowchart of Figure 4 is too small to see clearly.
  3. The format of the three tables in the article is not uniform.
  4. The color scheme of Figure 6 needs to be modified.

Author Response

Dear Sir or Madam,

Thank you for your careful reading, as well as the proposition of very important points for improving our work. Please find bellow a point-by-point response.

1. The resolution of Figure 1 is too low to give the reader a good look and feel.

R.: Thank you for your comment. Figure 1 has been modified and all figures was increased resolution.

2. The font in the flowchart of Figure 4 is too small to see clearly.

R.: Done. We totally agree with you.  We enlarged the font and reworded the text. Thanks.

3. The format of the three tables in the article is not uniform.

R.: Excellent observation, thank you. All tables have been adjusted and now are standardized.

4. The color scheme of Figure 6 needs to be modified.,

R.: The color scheme of figure 06 has been adjusted in accordance with the map legend of figure 12.

OBSERVATION: As we changed the title as suggested, we are thinking removing from Keywords (Line 35,36): Crop Livestock integration and iCL (As it is already mentioned in the new title) and adds training sample designs and samples balancing (As those are related to the developed methodology).  

Thank you very much